# TraceVLA: Visual Trace Prompting Enhances Spatial-Temporal Awareness for Generalist Robotic Policies

**Ruijie Zheng**[1][*][†]  **Yongyuan Liang**[1][*]  **Shuaiyi Huang**[1]
**Jianfeng Gao**[2]  **Hal Daumé III**[1]  **Andrey Kolobov**[2]  **Furong Huang**[1]  **Jianwei Yang**[2]
[1] University of Maryland, College Park
[2] Microsoft Research

## Abstract

Although large vision-language-action (VLA) models pretrained on extensive robot datasets offer promising generalist policies for robotic learning, they still struggle with spatial-temporal dynamics in interactive robotics, making them less effective in handling complex tasks, such as manipulation. In this work, we introduce *visual trace prompting*, a simple yet effective approach to facilitate VLA models' spatial-temporal awareness for action prediction by encoding state-action trajectories visually. We develop a new **TraceVLA** model by finetuning OpenVLA on our own collected dataset of 150K robot manipulation trajectories using visual trace prompting. Evaluations of **TraceVLA** across 137 configurations in SimplerEnv and 4 tasks on a physical WidowX robot demonstrate state-of-the-art performance, outperforming OpenVLA by 10% on SimplerEnv and 3.5x on real-robot tasks and exhibiting robust generalization across diverse embodiments and scenarios. To further validate the effectiveness and generality of our method, we present a compact VLA model based on 4B Phi-3-Vision, pretrained on the Open-X-Embodiment and finetuned on our dataset, rivals the 7B OpenVLA baseline while significantly improving inference efficiency.

## 1 Introduction

Robotic manipulation policies, typically trained on specific task demonstrations, often struggle to generalize beyond their training data, particularly when faced with novel objects, environments, instructions, and embodiments. In contrast, foundation models for vision and language—such as CLIP (Radford et al., 2021), LLaVA (Liu et al., 2024b), Phi-3-Vision (Abdin et al., 2024b), and GPT-4V (Achiam et al., 2023)—have demonstrated impressive generalization across diverse vision-language tasks. However, these models are not equipped to handle the challenges unique to robot manipulation, such as understanding kinematics, adapting to different embodiment configurations, and executing reliable physical actions. Vision-Language-Action models (VLAs) (Brohan et al., 2022; Kim et al., 2024) seek to address this gap by fine-tuning vision-language models to generate robot control actions using large-scale robotic datasets (e.g., Collaboration et al., 2023b), combining the generalization power of foundation models with task-specific robotic expertise. The approach has yielded promising results in developing generalist robot policies capable of adapting to a wide range of manipulation tasks.

However, VLA-powered robots often struggle to maintain awareness of their past movements, leading to decisions that are more reactive to current inputs rather than informed by spatial history. We posit that this limitation arises because simply mapping image inputs as current states to control actions is insufficient. To address this, we propose explicitly computing multi-point temporal trajectories and overlaying them directly onto the image inputs for VLA models. We hypothesize that this will effectively provide spatial and temporal relations necessary for improving manipulation tasks (Wen et al., 2023; Yuan et al., 2024).

---

[*]Equal contribution.
[†]Work done during internship at Microsoft Research.

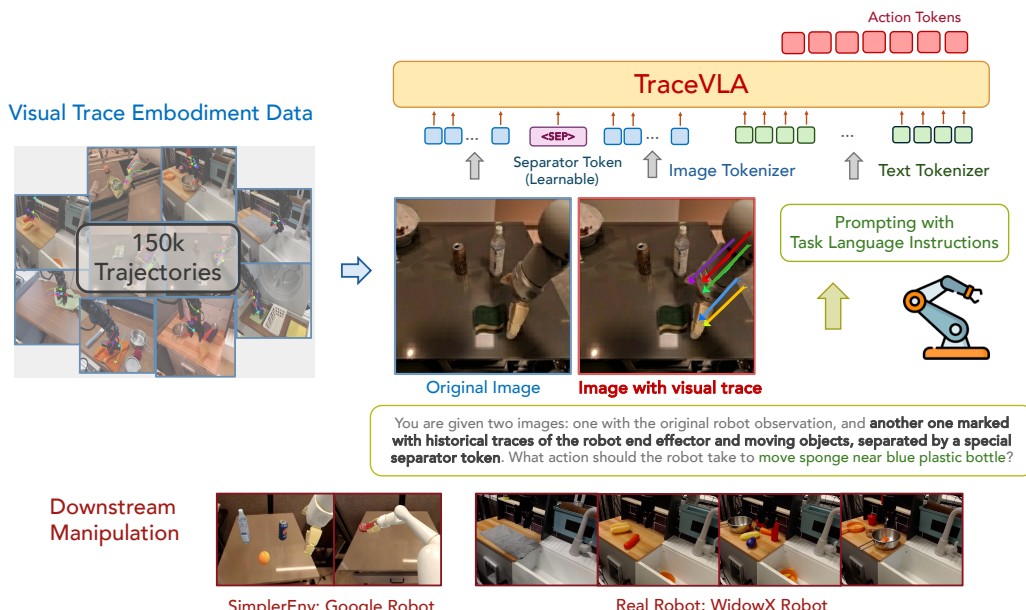

**Figure 1:** An illustration of our method. The first image shows the original robot's observation, while the second contains the same image with overlaid visual traces. A separator token is then inserted between the visual tokens of these two images, then concatenating with text tokens and feeding into the underlying vision language model backbone to output action tokens.

Our approach introduces an additional multi-point visual input during VLA training that tracks the robot's past movement trajectory (Karaev et al., 2023). We refer to these multi-point trajectories as *visual traces*, and show that even with only 2D images as inputs (which allows for better scalability and integration with existing VLM models), VLA models enhanced with visual traces exhibit improved spatial-temporal awareness. This visual trace prompting technique enables better adaptation to variations in manipulation tasks and improves overall generalization.

We introduce **TraceVLA**, a 7B-parameter VLA model fine-tuned from OpenVLA using our novel visual trace prompting dataset, which includes 150K robot manipulation trajectories as shown in Figure 1. In additional, we finetuned a more compact VLA model, **TraceVLA**-Phi3, using the 4B-parameter Phi-3-Vision as a backbone on the Open X-Embodiments dataset, which comprises 970K trajectories across diverse robot embodiments, tasks, and scenes. **TraceVLA**-Phi3 offers improved inference efficiency and reduced computational requirements while maintaining robust performance.

To assess the generalization capabilities of our models, we conducted evaluations across 131 diverse environment settings on Google Robots in the SimplerEnv simulator, which closely mimics real-robot scenarios. Additionally, we design diverse manipulation experiments on a physical WidowX robot to evaluate performance in real-world settings. Notably, our models consistently outperform existing VLA models across all embodiments and environments, demonstrating exceptional generalization under environmental variations.

Our key contributions are summarized as follows.

- **Method.** We introduce visual trace prompting, a novel technique that significantly enhances VLA models' spatial-temporal reasoning in manipulation tasks.

- **Dataset & models.** We provide a visual trace prompting dataset and present state-of-the-art 7B and 4B VLA models fine-tuned using our proposed visual trace prompting, offering an efficient method to boost VLA model performance.

- **Validation.** Our approach is rigorously validated through extensive evaluations in both simulated and real-world robot tasks across diverse embodiments, demonstrating superior generalization capabilities by leveraging spatial-temporal information.

## 2 PRELIMINARIES

**Visual-Language-Action Models.** Vision-language-action models (VLAs) extend vision-language models (VLMs) to predict discretized robot actions. Their architecture comprises: (1) a visual encoder (Zhai et al., 2023; Oquab et al., 2023) that converts input images into patch embeddings, (2) a projector that maps these embeddings to the language model's input space, and (3) a large language model backbone (Touvron et al., 2023). Action discretization, a crucial feature of VLAs, involves mapping continuous robot actions to discrete tokens. This process typically divides each action dimension into 256 bins based on data quantiles. These discrete actions are then incorporated into the language model's vocabulary, often replacing the least frequently used tokens. VLA training builds upon the foundation established during VLM training. During VLM training, the model is trained end-to-end with a next text token prediction objective on paired or interleaved vision and language data curated from various Internet sources. VLA training then extends this approach, encompassing fine-tuning of the pretrained VLM.

**Generalist Control Policies.** Robotic policy learning typically relies on task-specific demonstrations $\mathcal{D} = \{\tau_1, \tau_2, ..., \tau_n\}$, where each $\tau_i = \{(o_t, s_t, a_t)\}_{t=1}^T$ represents an expert-level trajectory. The learning architecture comprises a visual encoder $\mathcal{F}_\phi$, mapping image observations $o_i$ to features $z_i = \mathcal{F}_\phi(o_i)$, and a policy network $\pi_\theta$ outputting action distributions $\hat{a} \sim \pi_\theta(\cdot|z, s)$. Training minimizes the error between predicted $\hat{a}$ and optimal $a$ actions. To overcome task-specificity limitations, generalist policies are being developed, aiming to handle diverse sensors, action spaces, and robotic platforms in various scenarios. Vision-Language-Action (VLA) models, leveraging VLM's visual understanding and multimodal reasoning capabilities, show promise in creating more adaptable generalist policies. These models offer improved generalization across tasks, enhanced semantic understanding of environments, and the ability to follow natural language instructions, paving the way for more flexible and intuitive robotic control in broader applications.

## 3 TRACEVLA

Our ultimate goal is to equip the Vision-Language-Action (VLA) model with the necessary context to better understand both temporal and spatial dynamics. In this section, we describe the details of our method, **TraceVLA**, to achieve this objective. First, we introduce visual trace prompting in Section 3.1. Next, we explain the model architecture of **TraceVLA** in Section 3.2. Finally, we provide the implementation details in Section 3.3.

### 3.1 VISUAL TRACE PROMPTING

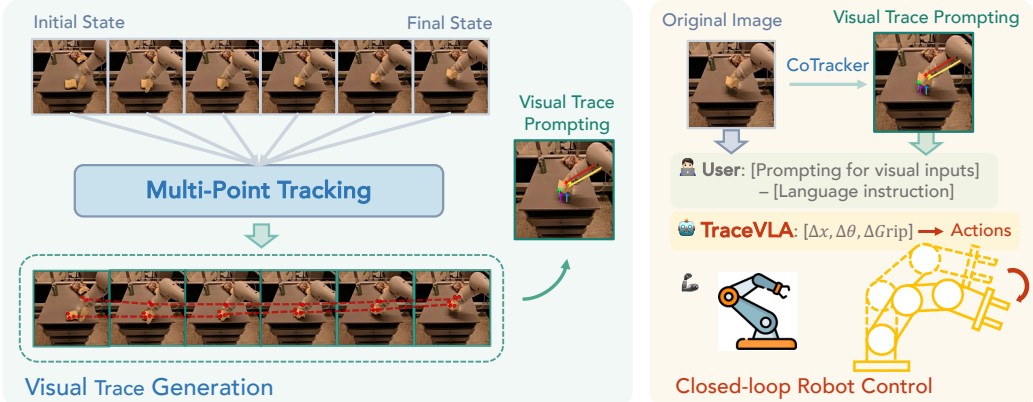

**Figure 2:** An illustration of visual trace generation. Given a sequence of historical image observations, we first use Co-tracker to extract dense point trajectories and keep active point trajectories with significant movement. We then overlay active point trajectories on the robot's initial observation frame as visual trace prompting. We feed both the image overlaid with visual traces and the original image into VLA as model input.

A straightforward way to have VLMs understand temporal history information is to concatenate historical frames and feed them into the VLA model. However, this approach can often distract the

model, as the frames are typically highly visually similar and redundant, making it difficult for the model to focus on the control-relevant information.

In this work, we introduce visual trace prompting to address this challenge. Instead of naively concatenating history frames, we employ an off-the-shelf point tracking algorithm to generate traces of key points. These visual traces are then visually overlaid on the robot's original observations, serving as visual prompts that provide the model with a spatial memory of its historical actions.

At any given timestep $t$ and a time window budget $N$, we first extract a set of dense point trajectories, $\mathcal{P}$, from a sequence of historical image observations, $\mathbf{h}_t = (o_{t-N}, \ldots, o_t)$, using a dense point tracking model. Specifically, a point trajectory represents the trace of a moving point over time. Thus, a set of dense point trajectories captures the traces of multiple critical moving points. The specific dense point tracking model we employ is Co-Tracker (Karaev et al., 2023), chosen for its efficiency and simplicity. Co-Tracker partitions the starting frame $o_t$ into a $K \times K$ grid and tracks each grid cell across $N$ frames to construct point trajectories. Consequently, we generate a total of $K \times K$ dense point trajectories, with each trajectory representing the location of a single point from timestep $t - N$ to timestep $t$.

While Co-Tracker provides a $K \times K$ grid of dense point trajectories, it does not inherently identify "active points." To address this, we identify active point trajectories in $\mathcal{P}$ by analyzing changes in pixel locations, focusing on those with significant movement and distinguishing them from static background points. For each point trajectory $\mathbf{p} \in \mathcal{P}$, we first compute the absolute movement $\Delta \mathbf{p}_{t'}$ between two adjacent frames at timestep $t'$ and $t' + 1$, as $\Delta \mathbf{p}_{t'} = \left| \mathbf{p}_{t'+1} - \mathbf{p}_{t'} \right|_1$. We then identify active point trajectories $\hat{\mathcal{P}}$ by computing the total movement over $N$ timesteps, keeping those whose movement exceeds a threshold $\kappa$. In other words, $\hat{\mathcal{P}} = \left\{ \mathbf{p} \in \mathcal{P} \mid \sum_{t'=t-N}^{t-1} \Delta \mathbf{p}_{t'} > \kappa \right\}$. From $\hat{\mathcal{P}}$, we randomly sample $M$ active point trajectories, denoted as $\tilde{\mathcal{P}}$, for use in visual prompting.

Finally, we generate the visual trace by overlaying the sampled active point trajectories $\tilde{\mathcal{P}}$ onto the robot's original observation frame $o_t$, as shown in Figure 2. This overlaid frame serves as a visual prompt, providing the model with spatial information about its historical states and actions.

## 3.2 MODEL ARCHITECTURE DESIGN

In Figure 1, we outline the design of TraceVLA using visual trace prompting. While the overlaid visual trace provides valuable spatial-temporal information about the robot's historical movements, it may obstruct the robot's end-effector or key objects, potentially hindering the model's ability to generate the correct action. To address this, we also include the original image observation in the model input, inserting a special separator token between the two images. As shown in Figure 1, we adjust the text prompt to inform the VLA model of this additional visual input before requesting the appropriate action output.

Additionally, since visual traces may not be available in all test-time scenarios—such as when the Co-Tracker model fails under pool lightning conditions—we implement a dropout mechanism during training. For each training example, with probability $\alpha$, we replace the visual trace prompt image with the original image and remove the corresponding hint from the text prompt. This dropout strategy introduces variability into the inputs, encouraging the model to effectively utilize both the original image and the visual trace. As a result, at test time, even if the Co-Tracker model is unable to extract the visual trace, our model can still function correctly.

## 3.3 IMPLEMENTATION DETAILS

In this section, we describe the implementation details of the dataset and models used in this work.

For the visual trace generation pipeline in training, we use a grid size of K=40, sample M=5 active point trajectories, and employ a time window N=6. To reduce computational overhead, we run dense point tracking every N step for the future 2N frames, rather than at every timestep. We divide demo trajectories into overlapping 2N-sized segments (e.g., [0, 2N], [N, 3N], etc.) and run Co-Tracker once per segment. This approach ensures that for each timestep t > N, at least N steps of historical context are available, while significantly reducing computational costs.

To create our dataset with visual trace annotations, we applied our visual trace generation pipeline to the following training datasets: BridgeData-v2 (Walke et al., 2023), Google RT1 Robot datasets (Brohan et al., 2022), as well as to 120 demonstrations collected from 4 manipulation tasks on our WidowX-250 Robot setup for downstream evaluation. In total, we gathered approximately **150,000** robot trajectories annotated with visual trace prompting, forming our fine-tuning dataset for TraceVLA.

To obtain real-time visual traces during inference, we reduce the computational overhead of densely querying the Co-tracker at every timestep by tracking M active points and sparsely querying Co-tracker. Specifically, at timestep t=0, we perform dense K×K point tracking to identify these active points (see Sec. 3.1 for details). For every timestep t>0, we query Co-tracker only for these active points to generate M corresponding traces, and update the tracked active points from the traces. For further details and pseudocode of our **TraceVLA** inference pipeline, please refer to Appendix E.

For the VLA models, we started with OpenVLA (Kim et al., 2024), a 7B VLA model based on the Prismatic vision-language model (Karamcheti et al., 2024), trained on the Open X-Embodiment dataset (Collaboration et al., 2023a). Additionally, we pretrained a 4B VLA model with Phi3-Vision as its backbone VLM (Abdin et al., 2024a), on the Open X-Embodiment dataset using a batch size of 4096 for 30 epochs with 32 H100 GPUs, following the same recipe as OpenVLA. This lightweight 4B model allows us to test the flexibility of our visual trace prompting across different VLM model architectures. Additionally, the 4B Phi3V-based VLA model will also provide the community with a more compact VLA model for finetuning compared to the larger 7B Prismatic model, while the reduced memory cost allows fine-tuning to be performed on smaller GPUs such as RTX4090 or RTX A5000's. For both **TraceVLA** and **TraceVLA**-Phi3, we finetune the base VLA model for an additional five epochs.

## 4 EXPERIMENT

To comprehensively evaluate our model's performance, we conducted experiments across a wide range of environmental setups, including 3 tasks with 137 different configurations in simulation and 4 tasks on real robots.

**Baseline.** We benchmark our approach against the following generalist policies, including state-of-the-art open-sourced models:

**OpenVLA** (Kim et al., 2024): A 7B parameter VLA trained on the Open-X-Embodiment (Collaboration et al., 2023a) Dataset, representing large-scale generalist policies.

**OpenVLA**-Phi3: A 4.5B parameter VLA pretrained model using Phi-3-Vision as backbone.

**Octo-Base** (Team et al., 2024): A 93M parameter transformer-based policy trained on 800k trajectories from the Open-X-Embodiment Dataset.

**RT1-X** (Collaboration et al., 2023a): A compact 35M parameter model trained on the same dataset as Octo-Base, exemplifying efficient architectures.

**TraceVLA** and **TraceVLA**-Phi3: Finetuned from OpenVLA and OpenVLA-Phi3 with visual trace prompting.

### 4.1 SIMULATION EVALUATION

**SimplerEnv.** Our simulation evaluation utilizes SimplerEnv, which incorporates two distinct settings: **visual matching** and **variant aggregation**. The visual matching setting aims to minimize the visual appearance gap between real environments and raw simulation, significantly enhancing the correlation between policy performance in simulation and real-world scenarios. Complementing this, the variant aggregation setting covers a wide range of environmental variations as shown in Figure 4, including backgrounds from different rooms, lighter and darker lighting conditions, varying numbers of distractors, solid color and complex table textures, and different robot camera poses. This comprehensive set of variations allows us to assess the robustness and adaptability of our approach in handling diverse manipulation scenarios, particularly evaluating the spatial and temporal awareness brought by visual trace prompting.

**Overall Performance.** As shown in Table 1, **TraceVLA** consistently outperforms OpenVLA across various tasks and evaluation metrics in the SimplerEnv Google robot tasks. The improvements are

| Models | Visual Matching | | | Variant Aggregation | | | Overall Performance |
|---|---|---|---|---|---|---|---|
| | Move Near | Pick Coke Can | Open/Close Drawer | Move Near | Pick Coke Can | Open/Close Drawer | |
| OpenVLA-Phi3 | 46.1% | 46.7% | 22.5% | 51.9% | 49.7% | 22.2% | 39.9% |
| **TraceVLA**-Phi3 | 50.4% (↑ **4.3**%) | 52.2% (↑ **5.5**%) | 31.0% (↑ **8.5**%) | 55.0% (↑ **3.1**%) | 52.4% (↑ **2.7**%) | 23.2% (↑ **1.0**%) | 44.0% (↑ **4.1**%) |
| OpenVLA | 47.1% | 15.3% | 49.5% | 54.0% | 52.8% | 22.5% | 40.2% |
| **TraceVLA** | 53.7% (↑ **6.6**%) | 28.0% (↑ **12.7**%) | **57.0**% (↑ **7.5**%) | **56.4**% (↑ **2.4**%) | **60.0**% (↑ **7.2**%) | 31.0% (↑ **8.5**%) | **47.7**% (↑ **7.5**%) |
| Octo-Base | 3.0% | 1.3% | 1.0% | 4.2% | 17.0% | 22.0% | 8.2% |
| RT1-X | **55.0**% | **52.8**% | 22.5% | 34.2% | 54.0% | **56.0**% | 45.8% |

**Table 1:** Performance results on three SimplerEnv Google robot tasks under two evaluation metrics: visual matching and variant aggregation. Overall performance is calculated as the average over all the results.

evident in both the full-scale 7B models (**TraceVLA** vs OpenVLA) and their 4B versions (**TraceVLA**-Phi3 vs OpenVLA-Phi3). **TraceVLA** shows significant performance gains, with improvements ranging from 2.4% to 12.7% in all three tasks across different metrics. When compared to other baselines like Octo-Base and RT1-X, both **TraceVLA** and **TraceVLA**-Phi3 generally perform better, with a few exceptions where RT1-X, shows competitive performance in specific tasks. These results suggest that the visual trace prompting technique employed in **TraceVLA** enhances the model's ability to generalize across different robotic manipulation tasks and environmental conditions, leading to improved performance in both visual matching and variant aggregation scenarios.

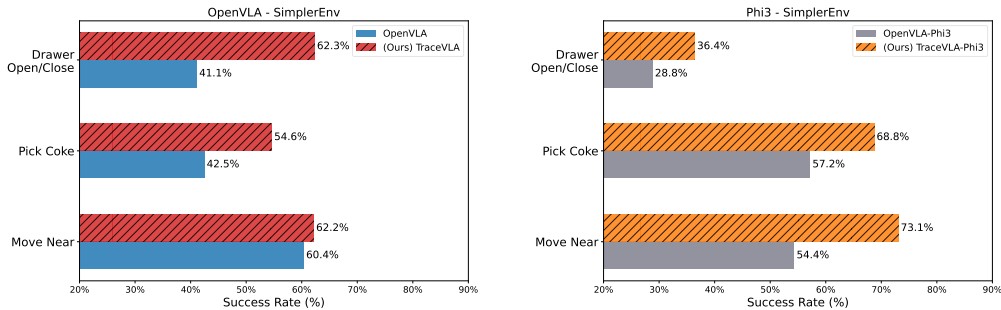

**Figure 3:** (**Left**): 7B **TraceVLA** vs. 7B OpenVLA. (**Right**): 4B **TraceVLA**-Phi3 vs. 4B OpenVLA-Phi3. Numbers are averaged across the visual matching and variant aggregation metrics.

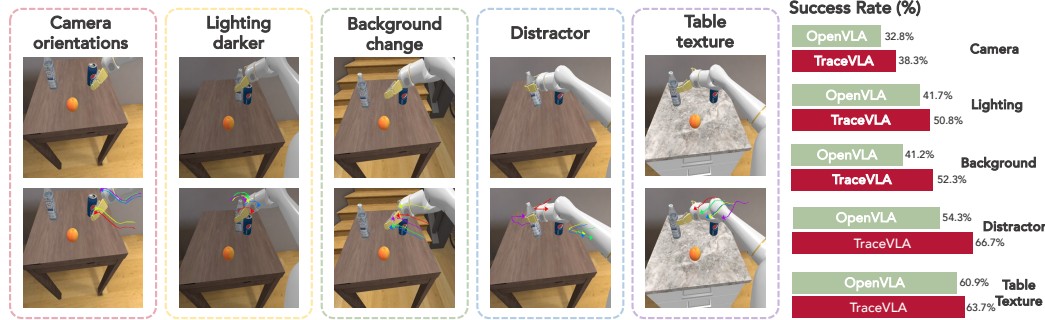

**Figure 4:** Comparison of OpenVLA and **TraceVLA** performance across various environmental variations: camera orientations, lighting, background, distractors, and table texture.

**Environmental Variant Aggregation.** Figure 4 demonstrates significant performance improvements of TraceVLA over OpenVLA across various environmental conditions. Notably, TraceVLA shows substantial enhancements under camera orientation changes, distractor presence, and background alterations, with an average improvement exceeding 20% in these categories. The use of visual trace prompting proves particularly effective when camera angles shift, as the trace provides valuable spatial trajectory information. This helps the model maintain performance despite changes in perspective.

Similarly, when faced with background changes, varying table textures, or lighting alterations, the visual trace allows the model to remain more stable and less influenced by environmental backgrounds. In scenarios with added distractors, the visual trace effectively reminds the policy of past trajectory interferences, evidenced by noticeable arm diversions. These varied environmental conditions collectively validate the spatial and temporal understanding provided by visual trace prompting.

## 4.2 REAL ROBOT EXPERIMENTS

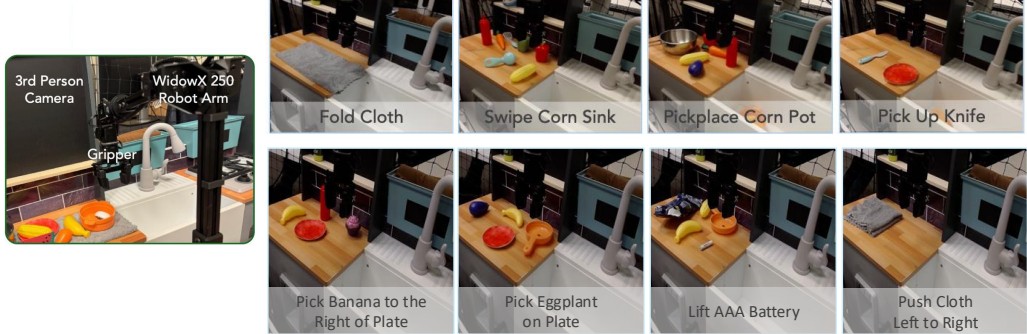

**Figure 5: Real robot setup.** We design 8 real-world robot tasks with different manipulation skills and objects including 4 unseen tasks for generalization evaluation.

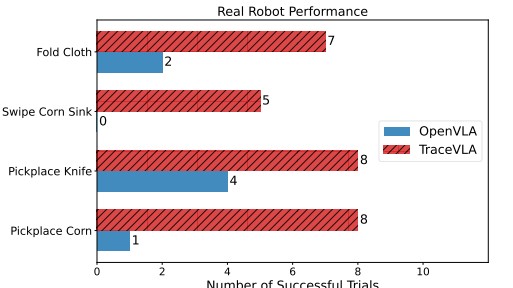

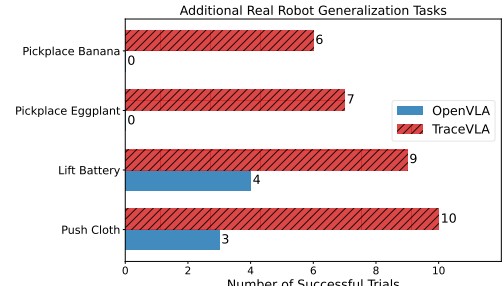

**(a)** **TraceVLA** outperforms OpenVLA on diverse real-robot manipulation tasks.

**(b)** **TraceVLA** showcases superior generalization on unseen real robot experiments.

**Figure 6:** Performance comparison of **TraceVLA** and OpenVLA on 8 real-world WidowX-250 robot manipulation tasks.

We evaluate TraceVLA on physical WidowX-250 robot manipulation tasks using a fixed-mounted third-person view camera capturing $256 \times 256$ RGB images. Despite sharing the same robot embodiment as BridgeData-v2, differences in setup, lighting, and camera angles necessitated collecting 30 demonstration trajectories per task for finetuning. Detailed real-robot experimental setup and protocols are in Appendix A.

As shown in Figure 6a, TraceVLA consistently outperforms the baseline across diverse tasks including soft object manipulation, pick-and-place operations, and object movement. Notably, in the pick-place corn task, which was not included in training data, TraceVLA achieved 8/10 successful trials compared to OpenVLA's 1/10, demonstrating strong generalization from similar training tasks, e.g., picking and placing eggplant in a pot.

To further evaluate generalization capabilities, we conducted additional experiments with four unseen tasks involving novel objects, goals, and language instructions. These trials included 2-3 random distracting objects in the scene except for pushing cloth. The results in Figure 6b demonstrate TraceVLA's superior generalization over OpenVLA. In the pick-place banana task, TraceVLA's only failures occurred due to grasping issues, while OpenVLA, even when successfully grasping the banana, failed to follow the language instruction by placing it on the plate rather than to its right. This highlights TraceVLA's enhanced language grounding capability and resistance to spurious correlations.

## 4.3 ABLATION STUDIES

To analyze the performance gain from visual trace prompting, we further study the following questions.

**Is performance gain of TraceVLA coming from further finetuning on a smaller subset of Open X-embodiment dataset?** To answer this, we also tested the performance of the 7B OpenVLA and 4B OpenVLA-Phi3 models finetuned on the exact same dataset as ours, but without using visual trace prompting. The results, as shown in Figure 7 (**Left**), indicate that the performance gain observed in TraceVLA is unlikely due to finetuning the pretrained VLA model on a smaller subset. While finetuning the pretrained OpenVLA model brings a 1.1% gain, finetuning OpenVLA-Phi3 model even degrades the performance by 0.3%. However, when visual trace prompting is incorporated, the success rate of OpenVLA model increases to 47.7%, highlighting the significant impact of visual traces on model performance.

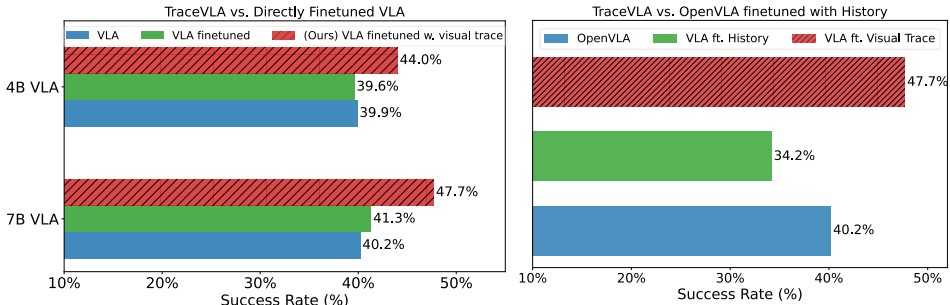

**Figure 7:** (**Left**): Comparison of average success rates between the base OpenVLA and OpenVLA-Phi3 models and their finetuned versions, with and without visual trace prompting. (**Right**): Comparison of average success rates between the base OpenVLA,**TraceVLA**, and OpenVLA finetuned with a sequence of 6 images.

**Will appending historical image observations also give similar performance gain as TraceVLA?** Instead of using visual trace prompting, the most straightforward way to integrate the model's historical movement knowledge is to provide it with multiple past image observations. To test this method, we input $N$ frames into the VLA model, separated by a learnable separator token, similar to our approach in **TraceVLA**. We also include a sentence in the text prompt to inform the model of this change in observation. Here, $N = 6$, which matches the length of the visual trace used in **TraceVLA**. In theory, the model should receive more information from these 6 complete frames compared to visual trace prompting. However, as shown in 7 (**Right**), finetuning OpenVLA with historical information not only fails to improve overall performance but also reduces it by 6%. This performance drop is likely due to redundant information between visual tokens at different timesteps, which may distract the model from focusing on the most relevant information for deciding next actions. In contrast, visual trace prompting offers useful hints that enhance the context for the vision-language model.

**Is visual trace prompting better than text trace prompting for grounding VLA with temporal-spatial understanding?** In addition to guiding the model with point tracking by overlaying the visual trace on the original image, we can also describe the movement of the points using 2D coordinates in text. To assess whether visual prompting is the most effective method for improving the VLA model's spatial-temporal understanding, we implemented an alternative approach that describes the points' movement verbally, treating the trace as text tokens, as shown in Figure 8 (**Right**).

Interestingly, using this text-based trace yields a 2.4% average performance gain over the baseline VLA model, suggesting that point tracking information is indeed useful for the robot model. However, compared to our approach (**TraceVLA**), the additional 6.4% gain indicates that leveraging the vision encoder's ability to process visual prompts is much more effective than describing the points' location and movement in text. While text descriptions can precisely convey the location and movement of each point, they also increase token count (by ∼150 tokens) compared with visual trace prompting, leading to substantially higher GPU costs. Moreover, relying solely on text fails to fully leverage the multimodal grounding capabilities of current vision-language models. Our visual trace prompting approach strikes an optimal balance between efficiency and efficacy, demonstrating superior performance gains (an additional 6.4% improvement) over text-based alternatives while maintaining computational efficiency.

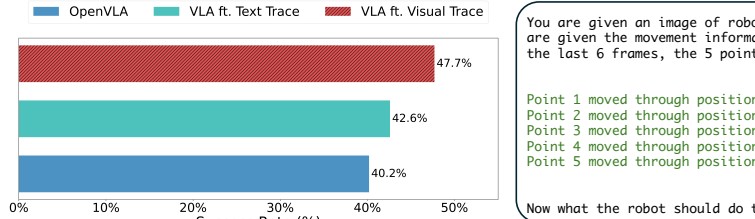

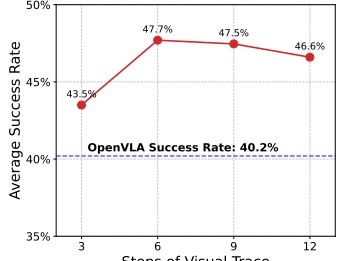

**Figure 8:** (**Left**): Comparing visual trace prompting and text trace prompting. (**Right**) Text trace prompts example.

**How does TraceVLA perform under different lengths of visual trace?** An important hyperparameter of **TraceVLA** is the length of visual trace prompting $N$, which determines the length of the historical information provided in the visual prompt. While a longer $N$ includes more past observations, it can clutter the visual context and potentially obscure key objects or the robot end-effector, while a shorter $N$ has less risk of covering the key information but contains less historical information. As shown in Figure 9, using a smaller number of steps ($N = 3$) results in a 3.2% performance improvement.

**Figure 9: TraceVLA** under different length of visual traces.

However, since a visual trace of only three steps is often too brief, the performance gain is not as substantial compared to larger values of $N$. Conversely, setting $N$ too large, such as $N = 12$, may lead to a less informative trace that obscures key scene elements and overlaps partially with previous motion traces of robots in the overlaid images, potentially distracting the VLM model, thereby slightly reducing overall performance. In practice, tuning $N$ should be straightforward, and might be dataset-specific. It involves sampling a few episodes and visually inspecting the generated trace to ensure that the selected $N$ provides an appropriate balance between historical context and the clarity of the scene.

## 5 LIMITATION ANALYSIS: TRAINING MEMORY COST AND INFERENCE SPEED

Since **TraceVLA** introduces an additional image input into the model and uses CoTracker to obtain the visual trace during testing, we examine both the training memory cost and inference speed of **TraceVLA**. For evaluating memory cost, we launch a single-node multi-gpu training job with 8 H100 graphics cards under varying batch sizes, and we measure the maximum GPU memory usage across 8 H100s. All tests are conducted with flash attention and torch.bfloat16 as the datatype of model weights and inputs. As shown in Figure 10 (left), when the batch size is 32, the memory difference between **TraceVLA** and models without visual trace prompting (for both Phi3 and OpenVLA) is less than 10GB. Notably, this difference becomes even smaller with a reduced batch size, indicating that while **TraceVLA** incurs some extra GPU memory cost, this additional GPU memory cost remains manageable and not significantly impactful.

For testing inference speed, we evaluate the time cost using a single H100 GPU. In addition, we also analyze the time cost of each additional component introduced in **TraceVLA** compared to the original OpenVLA model. During inference time, the extra computation introduced by **TraceVLA** consists of approximately 300 additional image and text tokens for the transformer model at each timestep, 5-point CoTracker point tracking for every timestep, and $K \times K$ (where $K = 40$) dense point tracking every 20 steps. For the $40 \times 40$ dense point tracking, as it requires recalculation only every 20 steps, the average time cost per timestep is computed by evenly distributing the total cost over each timestep. As shown in Figure 10 (right), we observe that the additional text and image tokens result in a negligible inference cost (around 0.002 seconds), likely due to GPU optimization for attention. Using $M$ points ($M = 5$) for CoTracker per timestep adds an extra 0.03 seconds per step, while the dense $40 \times 40$ point tracking has an amortized cost of 0.004 seconds per step.

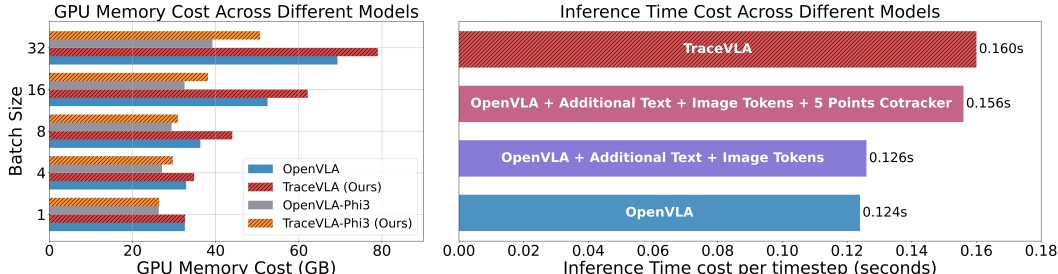

**Figure 10: (Left)**:Comparison of GPU memory cost of 7B **TraceVLA**, OpenVLA and 4B **TraceVLA**-Phi3, OpenVLA-Phi3. **(Right)**: Comparison of inference time across different models.

## 6   RELATED WORK

**Vision Language Action Models.**   Several studies have investigated the application of vision-language models (VLMs) in robotics (Karamcheti et al., 2023; Gadre et al., 2022; Driess et al., 2023; Du et al., 2023). Among them, Robopoint (Yuan et al., 2024) and RepKep (Huang et al., 2024b) leverages VLM for explicit key point coordinates prediction, which is then converted to low-level actions through an off-the-shelf motion planner. Meanwhile, many recent works have explored fine-tuning large pretrained VLMs to directly predict robot actions as VLA models, treating these actions as tokens within the language model vocabulary (Brohan et al., 2023; Niu et al., 2024; Zhu et al., 2024; Li et al., 2024; Kim et al., 2024). Among them, RT-2 (Brohan et al., 2023) fine-tuned VLMs on both robotic trajectory data and Internet-scale vision-language data. LLARVA (Niu et al., 2024) generated both 2D visual traces in image coordinates and corresponding textual actions as outputs, with the former functioning as an auxiliary task. LLaRA (Li et al., 2024) generates multiple auxiliary datasets with complementary training objectives to provide additional supervision. RT-2-X (Collaboration et al., 2023a) trains a 55B-parameter VLA policy on the Open X-Embodiment dataset. OpenVLA (Kim et al., 2024) combines a open VLM backbone with a richer robot pertaining dataset. We build on top of OpenVLA, but distinctively address the challenge of maintaining awareness of past spatial trajectories in VLA.

**Visual Trace for Robotics.** Visual traces of moving objects are vital for improving robotic action prediction, as they convey essential information about object dynamics. Various approaches have been developed to utilize visual traces in robotics, including using hand-drawn sketches for goal specifications (Gu et al., 2023), predicting future traces and learning a trace-guided policy (Wen et al., 2023; Bharadhwaj et al.), identifying active points for motion planning (Vecerik et al., 2024), and localizing active regions of robot observations for video generation (Huang et al., 2024a). More recently, the vision language action model LLARVA (Niu et al., 2024) predicts future 2D traces in text format as intermediate outputs alongside action tokens. In contrast, our approach integrates historical visual traces directly into VLA models as visual prompts. This novel method enhances VLMs' contextual understanding of the spatial and temporal dynamics, addressing an aspect previously underexplored in VLA models.

## 7   CONCLUSION AND DISCUSSION

Our work advances vision-language-action (VLA) models for robotic manipulation by introducing a novel visual trace prompting technique and providing a dataset enriched with spatial-temporal information across diverse embodiments. With state-of-the-art 7B and 4B VLA models, we push the boundaries of VLA performance, demonstrating their effectiveness in both extensive simulated environments and real-world robotic tasks. By bridging the gap between visual perception, temporal awareness, and physical embodiment, we significantly enhance the generalization and adaptability of VLA models.

Looking ahead, promising directions for future research include incorporating multi-point spatial trajectory prediction, allowing models not only to react but also to anticipate and plan actions with greater foresight. Additionally, leveraging 3D point cloud data for training could further enrich spatial representations, capturing fine-grained details in complex scenes and objects, thus improving manipulation accuracy and robustness across diverse and dynamic scenarios. These advancements will continue to enhance the generalization capabilities of VLA models, driving further progress in robotic manipulation.

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

## A    REAL ROBOT TASKS SETUP

Here we provide the detailed language instruction for each task that we designed. For each trial, we randomize the intial location of the target object, and for each trial except for folding cloth, we have also added 2-3 random distracting objects into the scene, including toy pepper, eggplant, ketchup, carrot, and donut.

Tasks included in the finetuning dataset.

- **Task 1: fold cloth**: *Fold the cloth from right to left*. The trial is counted as a success only when the robot succeesfully grasp the right edge of the cloth and fold it to the left.

- **Task 2: swipe corn sink**: *Pick up the brush and then use the brush to sweep the corn into the sink, while avoiding collision with other objects.* The trial is counted as success only when robot successfully swipe the corn into the sink without colliding into existing objects on the table.

- **Task 3: pickplace corn pot**: *Pick up the corn and then put it into the pot*. The trial is counted as success only when the robot correctly picks up the corn and place into the pot. Note that the primary goal of the this task is to assess the model's generalization capability, as this task is not part of the training dataset. Instead, the training data includes a task that involves picking up an eggplant and placing it in a pot.

- **Task 4: pickup knife**: *Pick up the knife first, and then place it on the plate*. The trial is counted as success only when the robot correctly picks up the knife and place into the target plate.

Unseen tasks for generalization:

- **Task 1: pickplace banana**: *Pick up the banana and place it to the right of the plate*. The trial is counted as a success only when the robot correctly picks up the banana and places it to the right of the plate. This task is particularly challenging because the banana object is unseen in our real-robot finetuning dataset. Additionally, solving this task requires the model to leverage its language understanding capability to ground spatial knowledge, rather than relying on spurious correlations, as the instructions in our finetuning dataset only involve placing objects on the plate.

- **Task 2: pickplace eggplant**: *Pick up the eggplant and place it on the plate*. The trial is counted as a success only when the robot correctly places the eggplant on the plate. This task tests the model's capability for handling unseen goals, as the finetuning dataset only includes placing the eggplant into a pot. Additionally, the eggplant is difficult to grasp, as incorrect placement of the end-effector could easily cause the eggplant to rotate and miss the target.

- **Task 3: lift battery**: *Lift the AAA battery*. The trial is counted as a success only when the robot correctly picks up the battery and lifts it without dropping or damaging it. This task tests the model's capability to handle unseen objects, as the battery is not included in our finetuning dataset.

- **Task 4: push cloth**: *Push the cloth from the left to the right of the table*. The trial is counted as a success only when the robot successfully pushes the cloth to within 1 inch of the right edge of the table. This task evaluates the model's motion generalization capability, as the finetuning dataset only includes tasks involving folding cloth.

# B  QUALITATIVE RESULTS ON REAL ROBOT ROLLOUTS

In this section, we present real robot manipulation rollouts for both the OpenVLA and **TraceVLA** models. As discussed earlier, our **TraceVLA** model demonstrates significantly better generalization ability across various real robot manipulation tasks, unseen objects, and unseen language instructions. In Figures 11, 12, and 13, we qualitatively illustrate how the two models handle three tasks: "Pickplace Banana, Folding Cloth, and Pickplace Eggplant." For the **TraceVLA** model, we also visualize the visual trace prompt that the model uses during evaluation.

Due to the proposed visual trace prompting, our **TraceVLA** model not only accurately picks up the banana and eggplant, grasps the edge of the folding cloth, and completes these tasks smoothly, but also demonstrates superior spatial understanding and reasoning capability compared to OpenVLA. In contrast, the OpenVLA model shows limited generalization capability, often overfitting to the finetuning distribution. For example, it places the banana directly onto the plate instead of following the instruction to place it to the right of the plate. These results further highlight the benefits of our visual trace prompting technique.

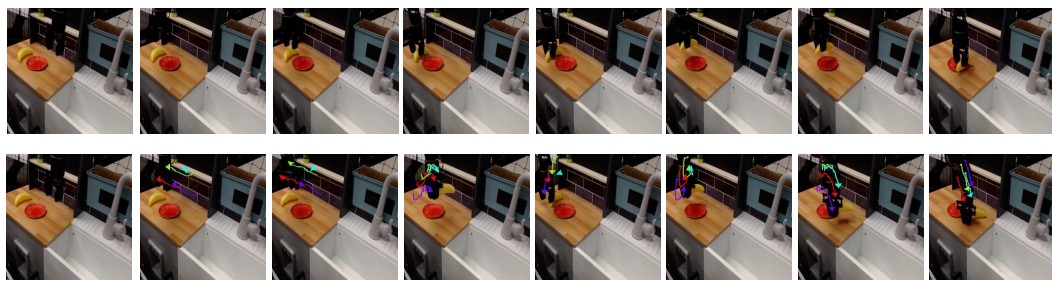

**Figure 11:** Pickplace Banana task. (Above): OpenVLA rollout. (Below): **TraceVLA** rollout with visual trace prompting.

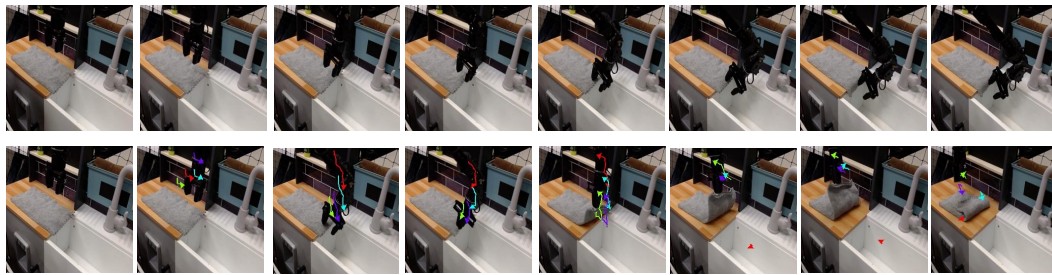

**Figure 12:** Fold Cloth task. (Above): OpenVLA rollout. (Below): **TraceVLA** rollout with visual trace prompting.

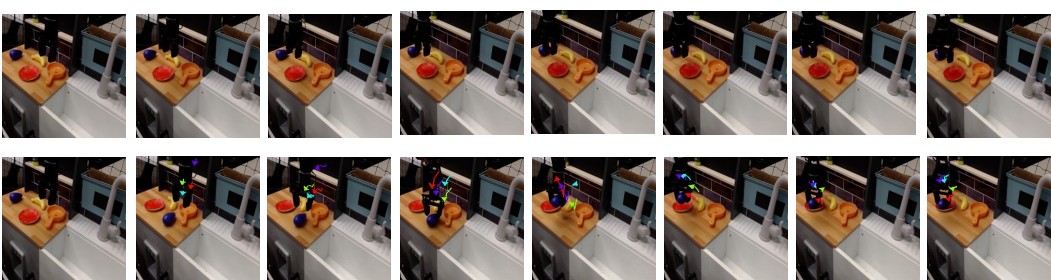

**Figure 13:** Pickplace Eggplant task. (Above): OpenVLA rollout. (Below): **TraceVLA** rollout with visual trace prompting.

## C    ADDITIONAL ABLATION STUDIES

### C.1    THICKNESS, TRANSPARENCY, AND COLOR OF VISUAL PROMPTING

To further investigate the impact of visualization parameters, we conducted additional ablation studies. Specifically, we fine-tuned the TraceVLA model on datasets with variations in trace visualization settings, including different line thicknesses, transparency levels, and color schemes. Our findings indicate that performance variations across these parameters are minimal within a reasonable range. Below, we provide detailed experimental results for each parameter setting.

**Thickness:** The effect of varying the line thickness of visual traces on the SimplerEnv Average Success Rate is shown in Table 2. We observe only minor differences in performance when adjusting this parameter.

| Thickness | SimplerEnv Average Success Rate |
|---|---|
| `linewidth=1` | 47.2% |
| **`linewidth=2` (TraceVLA)** | **47.7%** |
| `linewidth=3` | 47.8% |

**Table 2:** Impact of line thickness on performance.

**Transparency:** We varied the transparency of the visual traces by adjusting the $\alpha$ parameter. Lower $\alpha$ values result in more transparent traces. Table 3 summarizes the findings, demonstrating the robustness of TraceVLA's performance to these adjustments.

| Transparency ($\alpha$) | SimplerEnv Average Success Rate |
|---|---|
| $\alpha = 1$ **(TraceVLA)** | **47.7%** |
| $\alpha = 0.8$ | 47.3% |

**Table 3:** Impact of transparency on performance.

**Color:** The choice of color scheme was also tested. The default TraceVLA color scheme uses RYPBG (Red, Yellow, Purple, Blue, Green), while an alternative scheme POBGG (Pink, Orange, Blue, Grey, Green) was evaluated. Results are presented in Table 4, showing negligible differences in success rates.

| Color Scheme | SimplerEnv Average Success Rate |
|---|---|
| RYPBG (TraceVLA) | **47.7%** |
| POBGG | 47.3% |

**Table 4:** Impact of color scheme on performance.

**Takeaway:** Our experiments reveal that the choice of visualization parameters, including thickness, transparency, and color, has a negligible impact on TraceVLA's performance when chosen within reasonable ranges. These results suggest that such parameters do not require extensive hyperparameter tuning, simplifying their selection process. This robustness underscores TraceVLA's reliability across different visualization settings.

### C.2    BASELINE WITH DIFFERENT STEPS OF HISTORICAL OBSERVATIONS

In Figure 7, we compared TraceVLA with OpenVLA using 6 steps of observation history to ensure both models had access to the same amount of historical information. Here, in Figure 14, we further compare TraceVLA with OpenVLA fine-tuned using 2 and 3 steps of observation history on SimplerEnv. While a slight performance improvement is observed with 2-step history for OpenVLA, TraceVLA consistently and significantly outperforms the baseline in success rates, highlighting the effectiveness of visual trace prompting.

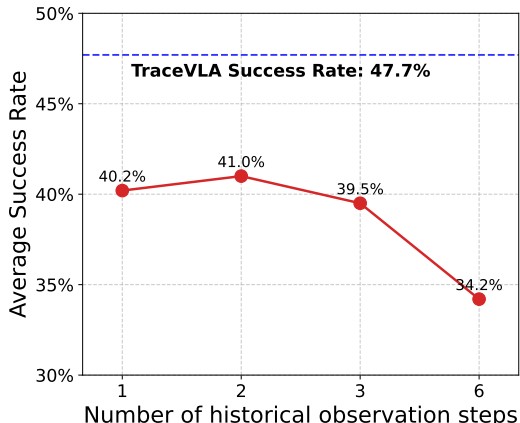

**Figure 14:** Comparison of TraceVLA against OpenVLA with different steps of observation history.

# D  ADDITIONAL RELATED WORK

In this section, we discuss some additional works that apply visual prompting technique of VLM and their applications in robotics. In particular, visual prompting methods have emerged as a new paradigm for VLM, complementing textual prompting and enabling more fine-grained and pixel-level instructions on multimodal input for VLMs (Yang et al., 2023a;b), and has been widely used in robotics (Yan et al., 2023; Liu et al., 2024a; Nasiriany et al., 2024). MOKA (Liu et al., 2024a) annotates key points as visual marks on images, converting affordance reasoning into a series of visual question-answering problems that are solvable by the VLM. PIVOT (Nasiriany et al., 2024) cast robotic control tasks as visual question-answering problems and iteratively refined visual prompts and action selection. Unlike existing work, our approach introduces visual trace prompting during fine-tuning of VLMs, overlaying key point traces on robot observations. Our novel visual trace prompting directly incorporates temporal information into the visual input, enhancing VLA models' spatial-temporal awareness for more effective action prediction in robotic tasks.

## E  MORE IMPLEMENTATION DETAILS

During inference, we aim to make our visual trace prompting as efficient as possible, adding minimal computation to the original VLA model. Extracting the visual trace by querying Co-Tracker for a $K \times K$ grid at each timestep is not feasible due to efficiency constraints. Instead, if we know the active points from the previous timestep, we can query Co-Tracker for only $M$ active points, which is faster and more cost-effective.

Ideally, similar to KV caching in LLM inference, we only run Co-Tracker with the $K \times K$ grid once at the start of the trajectory to find the $M$ active points. After that, we query Co-Tracker only for these $M$ active points throughout the trajectory. However, in practice, we observe that Co-Tracker might lose track after some steps (around 30 to 40, depending on the actions' magnitude). To address this, TraceVLA periodically re-queries Co-Tracker to recalibrate after a long interval. This ensures that the need for dense $K \times K$ point tracking is infrequent within an episode. As a result, the total number of dense queries during a trajectory is minimized, while tracking a few active points incurs little additional cost, adding minimal computational overhead to the model.

We refer the readers to Algorithm 1 for the pseudocode of TraceVLA model inference.

---

**Algorithm 1** Python-style pseudocode for TraceVLA Inference.

---

```python
# K: Co-Tracker Grid Size (e.g., 40 x 40)
# M: Number of Points to Track (e.g., 5)
# N: Trace Length for Co-Tracker (e.g., 6)
# T: Maximum timesteps for inference (e.g., 500)
# redraw_steps: Number of steps for Recomputing the KxK dense point tracking

# Initialization
image = env.reset()
historical_observations_queue = Queue(max_length=N)
tracked_points = None

for t in range(0, T):
    if t >= N:
        # KxK dense point tracking at timestep N or every redraw_steps for avoiding losing tracks
        if t == N or (t % redraw_steps == 0 and t > 0):
            # Recalculate K x K dense CoTracker point tracking
            grid_points = generate_grid_points(K, image.shape)   # Get K x K grid points
            trace = cotracker(historical_observations_queue, grid_points)
            trace = sample(trace, M) # Samplpe M visual traces (2 x N x M)
            # Update tracked points by using active points on the latest frame
            tracked_points = trace[:, -1, :]
        else:
            # Continue tracking with existing points
            trace = cotracker(historical_observations_queue, tracked_points) # (2 x N x M)
            # Update tracked points by using active points on the latest frame
            tracked_points = trace[:, -1, :]

        # Overlay trace on image and compute action using visual language model
        image_overlaid = overlay_trace(image, trace)
        action = traceVLA([image, image_overlaid], trace_prompt)

    else:
        # Use the prompt without trace hint for initial timesteps
        action = traceVLA([image, image], prompt)
    image = env.step(action)
    historical_observations_queue.append(image)
```

---

## F  ADDITIONAL EXPERIMENTAL RESULTS ON LIBERO SIMULATION BENCHMARKS

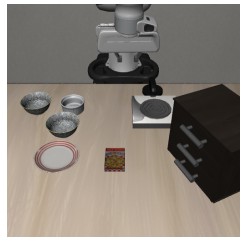 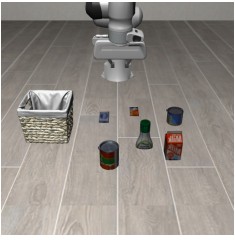 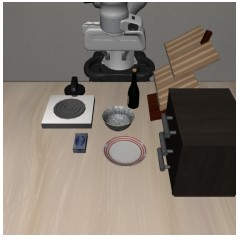 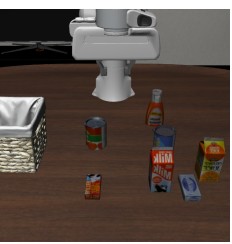

**(a)** LIBERO-Spatial    **(b)** LIBERO-Object    **(c)** LIBERO-Goal    **(d)** LIBERO-Long

**Figure 15:** Four test suites of LIBERO simulation benchmark.

| Method | LIBERO-Spatial | LIBERO-Object | LIBERO-Goal | LIBERO-Long | Average |
|---|---|---|---|---|---|
| **TraceVLA finetuned** | $84.6\% \pm 0.2\%$ | $85.2\% \pm 0.4\%$ | $75.1\% \pm 0.3\%$ | $54.1\% \pm 1.0\%$ | $74.8\% \pm 0.4\%$ |
| OpenVLA finetuned | $82.6\% \pm 0.4\%$ | $83.8\% \pm 0.6\%$ | $70.4\% \pm 0.5\%$ | $45.7\% \pm 0.6\%$ | $70.6\% \pm 0.4\%$ |

**Table 5:** Multitask success rates on LIBERO simulation benchmarks.

In addition to SimplerEnv and WIDOWX-250 real robot experiments, in this section, we conduct an additional experiment on LIBERO simulation benchmarks. In particular, we take the four suites from LIBERO: **LIBERO-long**, **LIBERO-Spatial**, **LIBERO-Object**, **LIBERO-Goal** in LIBERO, each with 10 tasks and 50 human teleoperated demonstrations per task. We evaluate the multitask performance of the pretrained VLA policy on these four suites.
Specifically:

- **LIBERO-Spatial**: Contains the same set of objects but in varying layouts, testing the model's ability to understand spatial relationships. Example language instruction: *pick up the black bowl between the plate and the ramekin and place it on the plate*.

- **LIBERO-Object**: Features consistent scene layouts but introduces different objects, evaluating the model's understanding of object types. Example language instruction: *pick up the alphabet soup and place it in the basket*.

- **LIBERO-Goal**: Maintains the same objects and layouts while varying task goals, assessing the model's knowledge of diverse task-oriented behaviors. Example language instruction: *put both the alphabet soup and the tomato sauce in the basket*.

- **LIBERO-Long** (also referred to as **LIBERO-10**): Comprises long-horizon tasks involving diverse objects, layouts, and task goals, challenging the model's ability to handle extended planning and execution. *Example language instruction: open the middle drawer of the cabinet*.

Following OpenVLA, we preprocess the data by filtering out non-successful trajectories and removing all steps with actions that have near-zero norms and do not change the gripper's status. For TraceVLA, we also annotate visual trace following the exact same procedure as what we described earlier, Bridge and Google Robot dataset. Then we finetune both the OpenVLA model and the TraceVLA-7B model on the combined dataset from these four suites and report their multitask success rates on each suite in table 5. (Note that compared to the numbers reported in the OpenVLA paper, here we finetune a single model on the mixture of all four suites altogether instead of finetuning on each suite separately and report the numbers.) For each task, we evaluate each method using three random seeds. For each seed, we perform 50 rollouts with random initial states and report the average performance across all 10 tasks per suite. As shown with table 5, compared with OpenVLA, the superior performance of TraceVLA across each benchmark suite further demonstrates benefits of our visual trace prompting.

