# OpenReview forum: "TraceVLA: Visual Trace Prompting Enhances Spatial-Temporal Awareness for Generalist Robotic Policies"
_ICLR.cc/2025/Conference — ICLR 2025 Poster_

### Official Review · Reviewer_qxt2 · 2024-10-30

**Soundness:** 3
**Presentation:** 4
**Contribution:** 3
**Rating:** 6
**Confidence:** 3

**Summary:**

This paper proposes overlaying the history trace on the RGB image as an indicator of spatial and temporal information to help large foundation models better manipulate objects. Experiments are conducted in both simulation and the real world.

**Strengths:**

Although trace has been implemented in various aspects, from my knowledge, this is the first time it serves as a history spatial-temporal information indicator. Comprehensive experiments and analyses are conducted to demonstrate its effectiveness, and the authors also consider real-time inference by optimizing the inference process.

**Weaknesses:**

1. The choice of active points depends on a threshold. Does this threshold need to be chosen differently for each robot or dataset?

2. In the analysis of trace length, the authors mention that a longer trace length can obscure key information. However, in the method, they state that they mitigate this by inputting both the original and overlay images. In that case, a longer trace length should not lead to worse results. This explanation seems self-contradictory."

3. For fine-tuning OpenVLA with history images, six history frames work best for TraceVLA, but this might be too redundant when using image. Would using three or two history frames be better when using history images as input?

**Questions:**

Does the dataset used for fine-tuning OpenVLA contain scenes similar to those in simplerEnv? That is, are all the testing scenes entirely unseen in the training dataset?

---

> ### Author Response · Authors · 2024-11-23
> **1/1 Response to Reviewer qxt2**
>
> Thank Reviewer qxt2 for the insightful feedback and for acknowledging TraceVLA's technical novelty and empirical contributions. We address your questions and concerns in detail below.
>
> > The choice of active points depends on a threshold. Does this threshold need to be chosen differently for each robot or dataset?
>
> We use a consistent threshold in CoTracker to track motions across different robot datasets during training, ensuring effective tracking without requiring dataset-specific tuning or embodiment-specific adjustments during inference. The threshold is chosen to strike a balance between precision and recall, ensuring that the selected active points are most indicative of motions relevant to manipulation tasks.
>
> In practice, setting this threshold is straightforward, as the movement magnitudes of active objects and static backgrounds generally differ significantly for the majority of trajectories. Any value within this range reliably separates points on the robot end-effector and active objects, making the approach robust across diverse datasets and embodiments.
>
> > The authors mention that a longer trace length can obscure key information. However, in the method, they state that they mitigate this by inputting both the original and overlay images. In that case, a longer trace length should not lead to worse results. This explanation seems self-contradictory.
>
> Thank you for this insightful question. Since we use 2D traces in the 2D image inputs, longer trajectories not only will obscure more observations, but more importantly may partially overlap previous motion traces of robots in the trace overlaid images, which could make the visual trace distracting and non-informative.
>
> > For fine-tuning OpenVLA with history images, six history frames work best for TraceVLA, but this might be too redundant when using image. Would using three or two history frames be better when using history images as input?
>
> We have conducted additional experiments fine-tuning OpenVLA with fewer history frames (2 or 3 frames), which we include in Figure 11 of Appendix B.2. While setting step=2 results in a slight performance gain (0.8%) over baseline OpenVLA, TraceVLA significantly outperforms these variants, demonstrating the effectiveness and efficiency of our trace representation in capturing temporal and spatial information.
>
> > Does the dataset used for fine-tuning OpenVLA contain scenes similar to those in simplerEnv? That is, are all the testing scenes entirely unseen in the training dataset?
>
> It is an important question about dataset composition. The tasks in SimplerEnv are not present in the fine-tuning dataset, but use the same robot embodiments (WidowX and Google Robot). We fine-tuned both models using the same real-robot dataset, since differences in the physical setup (like backgrounds, camera angles, and lighting conditions) make it difficult for any model to work immediately without training (as stated in line 338-343). Even with identical fine-tuning conditions, our model demonstrates stronger generalization capacity compared to baselines.
>
> To further evaluate generalization, we conducted additional real-robot experiments with four unseen tasks, specifically designed to test adaptation to novel objects, goals, language instructions, and motion scenarios. As shown in Figure 12, TraceVLA significantly outperforms baselines in these challenging scenarios, exhibiting stronger language grounding capability and generalizes beyond spurious correlations.
>
> We appreciate the constructive feedback that has helped improve our paper's quality. Additional results will be included in the Appendix. We sincerely hope our responses have adequately addressed your concerns and look forward to any further discussion.

---

> > ### Comment · Reviewer_qxt2 · 2024-11-25
> >
> > My concern has been addressed, I would like to raise my score.

---

### Official Review · Reviewer_4XPZ · 2024-11-02

**Soundness:** 3
**Presentation:** 3
**Contribution:** 3
**Rating:** 6
**Confidence:** 4

**Summary:**

This paper introduces TraceVLA, a novel method which leverages visual traces to enhance the spatial-temporal awareness of VLA models.
Specifically, the VLA model takes as inputs two images: a plain image and an image overlaid with traces of a set of active points.
The traces are generated by CoTracker and the active points are selected from a grid of dense points based on the changes in pixel locations.
The state-action history can be effectively communicated to the model through the provided visual traces.
Experiment results show that the proposed method outperforms comparing baseline methods in the SimplerEnv simulation environment and four real-robot tasks.
Ablation studies show that using visual traces is a more effective method for conveying historical information than prompting with text traces or appending historical image observation.

**Strengths:**

The proposed method, TraceVLA, is a neat and effective approach for providing large VLA models with historical state-action information.
Visual traces help VLA models to maintain awareness of the robot movements and do not introduce redundant information compared to appending historical image observations.
Experiments in SimplerEnv show that TraceVLA outperforms multiple comparing baseline methods in overall performance.
The paper also evaluates on real-robot tasks to validate the proposed method in the real world.
Ablation studies offer good insights on how leveraging visual trace prompts compares to other prompting strategies.

**Weaknesses:**

1. The experiments in the real world contains only one task for generalization assessment. It would be better to evaluate the proposed method in more generalization settings, including novel backgrounds and more novel tasks. This would provide a more comprehensive insight on the generalization capabilities of the proposed method.

2. RT1-X, despite having a much small number of parameters, outperforms TraceVLA-Phi3 with 3B parameters in SimplerEnv. The advantage of TraceVLA with 7B parameters is also not substantial. Is it possible to provide more discussions on potential reasons? Additionally, it would be beneficial to include RT1-X in real-robot experiments for comparison. In particular, observing how these two methods perform in generalization settings would be informative.

**Questions:**

1. The proposed method leverages CoTracker to provide visual traces. Were any failure cases due to inaccurate tracking observed during experiments? How robust is TraceVLA in handling these inaccuracies?

2. As the active points are selected based on changes in pixel locations, will points on moving background objects be chosen as active points? And how does the proposed method perform when the active points are located on irrelevant background?

3. How many iterations were used for training during fine-tuning on the data in real-robot experiments? Is the data from the pre-training stage also included in fine-tuning as well?

---

> ### Author Response · Authors · 2024-11-23
> **1/2 Response to Reviewer 4XPZ**
>
> Thank Reviewer 4XPZ for the thoughtful feedback and valuable suggestions. We appreciate Reviewer 4XPZ's acknowledge of TraceVLA’s technical and empirical contributions. Below, we provide detailed responses to each concern and question.
>
> > It would be better to evaluate the proposed method in more generalization settings, including novel backgrounds and more novel tasks.
>
> Thank you for this valuable suggestion regarding generalization evaluation. In our real-robot experiments, we have already added randomized background distracting objects for each trial to test model's robustness under novel background. Additionally, to test more novel tasks, we have expanded our experiments in two key aspects:
> * We conducted additional real-robot experiments with four unseen tasks designed to test generalization across novel objects, goals, language instructions, and motion scenarios in Appendix C. As shown in Figure 12, TraceVLA demonstrates significantly better generalization compared to OpenVLA when handling these novel elements.
> * We added qualitative visualizations in Appendix D to provide detailed insights into our model's behavior across different generalization scenarios. These results further validate the effectiveness of our visual trace prompting technique.
>
> > RT1-X, despite having a much small number of parameters, outperforms TraceVLA-Phi3 with 3B parameters in SimplerEnv. The advantage of TraceVLA with 7B parameters is also not substantial. Is it possible to provide more discussions on potential reasons? Additionally, it would be beneficial to include RT1-X in real-robot experiments for comparison.
>
> Thank you for this insightful question. We conducted a detailed evaluation of RT1-X by fine-tuning it on our real-robot dataset using the officially released RT1-X training code.
>
> In our real-robot experiments, RT1-X exhibited the following performance: for in-domain tasks like Fold Cloth, RT1-X was able to detect and track objects but struggled to complete the full task successfully. For out-of-domain tasks, it failed entirely, resulting in a zero success rate across all tested tasks. We attribute RT1-X's weak generalization capability in real-world scenarios to two key technical limitations:
> * **Weaker Language Understanding**: RT1-X employs the Universal Sentence Encoder for language embeddings, which offers limited language comprehension. This often led to failures in correctly identifying target objects. For instance, in a pick-and-place task involving a banana, RT1-X misinterpreted the language instruction and attempted to grasp the plate instead of the banana.
> * **Limited Visual Feature Extraction**: Compared with TraceVLA and OpenVLA which uses a fused DINO-SIGLIP vision encoder, RT1-X uses an EfficientNet-based vision encoder, which demonstrates suboptimal visual feature extraction capabilities in standard computer vision benchmarks. This resulted in imprecise grasping and repetitive basic pick motions without task completion.
>
> These findings underscore the superior generalization of VLM-based approaches like TraceVLA, which leverage robust language and visual encoders for handling complex real-world scenarios.
>
>
>
> > The proposed method leverages CoTracker to provide visual traces. Were any failure cases due to inaccurate tracking observed during experiments? How robust is TraceVLA in handling these inaccuracies?
>
> Indeed during our experiments, we a few failure cases due to tracking inaccuracies, particularly in scenarios where the background and objects or the robot share similar colors, or in cases of partial occlusions. However, our method incorporates multiple mechanisms to handle such challenges:
>
> * Dual-input design: TraceVLA leverages both original images and trace overlays. This ensures that critical visual details from the original image remain intact, allowing the model to disregard irrelevant or inconsistent traces caused by tracking inaccuracies.
> * Multi-point tracking: By tracking multiple points simultaneously, we maximize the likelihood of maintaining robust tracking for the robot, even when some points on the background are mistakenly included.
> * Periodic reinitialization: We periodically restart tracking to ensure traces' quality and robustness, allowing the tracking to recover from errors caused by background motion.

---

> > ### Author Response · Authors · 2024-11-23
> > **2/2 Respone to Reviewer 4XPZ**
> >
> > > As the active points are selected based on changes in pixel locations, will points on moving background objects be chosen as active points? And how does the proposed method perform when the active points are located on irrelevant background?
> >
> > Active points on moving background objects may occasionally be selected, especially in cluttered scenes or when background objects exhibit significant motion. However, our large-scale training dataset also includes some instances where CoTracker mistakenly tracks points on background objects, allowing TraceVLA to learn to adapt to small background perturbations during training. Additionally, the dual-input design, multi-point tracking, and periodic reinitialization mechanisms we mentioned earlier also collectively enable TraceVLA to remain robust even in such challenging conditions.
> >
> >
> > > How many iterations were used for training during fine-tuning on the data in real-robot experiments? Is the data from the pre-training stage also included in fine-tuning as well?
> >
> > During real-robot experiments, we fine-tune the model for 10 epochs using only the task-specific trajectories collected in the real environment. Pre-training data is not included in the fine-tuning stage to avoid potential domain gaps and ensure the model adapts to real-world visual characteristics. This protocol is applied consistently across all methods for fair comparison.
> >
> > Thank you again for your review! We sincerely hope our responses have adequately addressed your concerns and look forward to any further discussion. Please let us know if you have any further questions. We are more than happy to address them.

---

> > > ### Comment · Reviewer_4XPZ · 2024-11-25
> > > **Response to Authors**
> > >
> > > Thanks for the detailed response. All my concerns have been addressed. The updated experiments in the Appendix showcases that the proposed method possesses strong generalization capabilities on handling different unseen settings. Overall, I raise the rating to 6.

---

### Official Review · Reviewer_Gs1B · 2024-11-02

**Soundness:** 3
**Presentation:** 3
**Contribution:** 3
**Rating:** 8
**Confidence:** 4

**Summary:**

This work leverages visual trace prompting to enhance VLA models' spatial-temporal awareness for action prediction by visually encoding state-action trajectories. The authors curated a dataset of 150K robot manipulation trajectories using visual trace prompting and fine-tuned it on OpenVLA.

**Strengths:**

- The paper introduces visual trace prompting, a novel technique that enhances spatial-temporal reasoning in VLA models for manipulation tasks by representing spatial-temporal relationships in robotic contexts.

- A curated visual trace prompting dataset was developed, with state-of-the-art 7B and 4B VLA models fine-tuned using this approach, providing an efficient method to improve VLA model performance.

- Their approach was rigorously validated through extensive evaluations across diverse simulated and real-world robot tasks, demonstrating superior generalization by leveraging spatial-temporal information.

**Weaknesses:**

- While the author presented results highlighting the importance of historical information from the trace, there were no results demonstrating spatial reasoning capabilities as claimed in the "spatial-temporal" contribution. One potential approach could involve fine-tuning a Vision-Language Model (VLM) for line-drawing tasks, connecting it to low-level policy, and leveraging the VLM independently for spatial reasoning, similar to the RoboPoint framework.

- The author relied entirely on real-world data to generate the trace, which may have introduced variability from environmental factors such as lighting. Generating trace data in simulation could mitigate these issues, potentially minimizing the sim-to-real gap. Additionally, evaluating performance would be easier with ground truth data generated from a platform like RLBench across diverse tasks.

- Generalization in the Vision-Language Alignment (VLA) could be a strong aspect of this approach, as the data captures a general representation of actions conditioned on language. It would be beneficial for the author to test generalization on simulation benchmarks, such as the Colosseum, to showcase its robustness.

**Questions:**

Please refer to weakness section for my questions. I would consider raising my rating if the weaknesses mentioned in my questions can be addressed. Overall, I find this to be a strong paper that offers a valuable perspective on training VLA models beyond simply adding more data.

---

> ### Author Response · Authors · 2024-11-23
> **1/2 Response to Reviewer Gs1B**
>
> Thank you for your thoughtful and detailed review! We are encouraged that you think our visual trace prompting technique is novel and our experimental result is solid with extensive empirical evaluations. Below we address you questions in details.
>
> > While the author presented results highlighting the importance of historical information from the trace, there were no results demonstrating spatial reasoning capabilities as claimed in the "spatial-temporal" contribution. One potential approach could involve fine-tuning a Vision-Language Model (VLM) for line-drawing tasks, connecting it to low-level policy, and leveraging the VLM independently for spatial reasoning, similar to the RoboPoint framework.
>
> Thank you for your insightful question. We believe the spatial reasoning capability of our approach is demonstrated through the visual grounding of the VLA model, which bridges 2D visual trace prompting with the 7-dimensional 3D action execution. This capability is quantitatively validated in the success rates reported for both simulated and real-world robotics benchmarks.
>
> We appreciate your suggestion of RoboPoint as relevant related work and have incorporated it into our related work section. While RoboPoint fine-tunes a Vision-Language Model (VLM) to predict 2D keypoints and uses depth maps and motion planners to execute actions, our method, directly fine-tunes the VLA model to produce action tokens. The core focus of our method lies in efficiently encoding historical movements to enhance spatial-temporal awareness, particularly in leveraging past visual and action information for improved decision-making. This emphasis distinguishes our approach from RoboPoint, whose primary goal is keypoint prediction combined with motion planning.
>
> While our current implementation does not explicitly include keypoint prediction, integrating such auxiliary tasks with our VLA framework could be an intriguing direction for future exploration. This could further enrich the spatial reasoning aspect of our approach by combining explicit keypoint reasoning with direct action token prediction.
>
>
> > The author relied entirely on real-world data to generate the trace, which may have introduced variability from environmental factors such as lighting. Generating trace data in simulation could mitigate these issues, potentially minimizing the sim-to-real gap.
>
> In this work, our primary focus is on learning a generalist robot policy directly from real-world data, where diversity and variability in the training process are viewed as beneficial for improving robustness. The use of SimplerEnv Eval serves solely to validate the performance of our trained policy on real-world data, as the exact physical setup of the Google Robot dataset cannot be replicated. Addressing the sim-to-real gap is not within the scope of this work, as our approach exclusively trains policies on real-world data, with SimplerEnv functioning as a controlled evaluation metric rather than a simulation-to-real bridging tool.

---

> > ### Author Response · Authors · 2024-11-23
> > **2/2 Response to Reviewer Gs1B**
> >
> > > Additionally, evaluating performance would be easier with ground truth data generated from a platform like RLBench across diverse tasks.
> >
> > Thanks for your suggestions. We have conducted an extra experiment on LIBERO simulation benchmarks, where we take four suites LIBERO-long, LIBERO-Spatial, LIBERO-Object, LIBERO-Goal in LIBERO, each with 10 tasks and 50 human teleoperated demonstration per task. Following OpenVLA, we first preprocess the data by filtering out non-succesful trajectories, and meanwhile removing all steps with actions that have near-zero norm and do not change the gripper's status. We then finetune both OpenVLA model and TraceVLA-7B model on the mixture of data from these four suites and report its multitask success rate on each suite.
> >
> >
> > | Method | LIBERO-Spatial | LIBERO-Object | LIBERO-Goal | LIBERO-Long | Average |
> > |--------|----------------|---------------|-------------|-------------|---------|
> > | **TraceVLA finetuned** | **84.6%** | **85.2%** | **75.1%** | **54.1%** | **74.8%** |
> > | OpenVLA finetuned | 82.6% | 83.8% | 70.4% | 45.7% |70.6% |
> >
> > (Note that compared to the number reported in OpenVLA paper, here we finetune one single model on the mixture of four suites altogether instead of finetuning on each four domain seperately and report the number.)
> >
> >
> >
> >
> > > Generalization in the Vision-Language Alignment (VLA) could be a strong aspect of this approach, as the data captures a general representation of actions conditioned on language. It would be beneficial for the author to test generalization on simulation benchmarks, such as the Colosseum, to showcase its robustness.
> >
> > We appreciate your suggestion regarding testing generalization on simulation benchmarks like the Colosseum. Simulation indeed provides a valuable platform for evaluating various aspects of generalization. However, we would like to emphasize that our work already addresses a range of generalization settings using SimplerEnv. These include variations in camera orientations, lighting, background, distracting objects, and table texture, as illustrated in Figure 4 of the main paper. Additionally, we have also tested on the LIBERO benchmark as shown above. We believe that these evaluations haveve already comprehensively demonstrated the robustness of the trained VLA policy across diverse scenarios.
> >
> >
> > Thank you again for your insightful review. We sincerely hope our responses have adequately addressed your concerns and look forward to any further discussion. Please let us know if you have any further questions. We are more than happy to address them.

---

> > > ### Comment · Reviewer_Gs1B · 2024-11-23
> > > **Response to author**
> > >
> > > Thank you for your detailed feedback on our paper.
> > >
> > > >While the author presented results highlighting the importance of historical information from the trace, there were no results demonstrating spatial reasoning capabilities as claimed in the "spatial-temporal" contribution. One potential approach could involve fine-tuning a Vision-Language Model (VLM) for line-drawing tasks, connecting it to low-level policy, and leveraging the VLM independently for spatial reasoning, similar to the RoboPoint framework.
> > >
> > > We appreciate your suggestion. As you noted, the end-to-end nature of VLA models inherently makes it challenging to disentangle specific contributions, such as spatial reasoning. To strengthen our claims about "spatial-temporal" capabilities, a potential next step could involve evaluating tasks explicitly requiring spatial reasoning, which might provide more direct evidence.
> > >
> > > >Additionally, evaluating performance would be easier with ground truth data generated from a platform like RLBench across diverse tasks.
> > >
> > > Thank you for highlighting this. We appreciate your acknowledgment of our experiments on LIBERO. Regarding the comparison between OpenVLA and TraceVLA, we will include standard deviations and conduct statistical significance tests (e.g., p-tests) in future revisions to provide more robust evidence of the results.

---

> > > > ### Author Response · Authors · 2024-11-23
> > > > **Response to Reviewer Gs1B**
> > > >
> > > > Dear Reviewer Gs1B,
> > > >
> > > > Thanks for your timely reply. But it seems that you provided some example responses for your original proposed questions. We did not see any feedback regarding our rebuttal. Is there anything unclear or requiring further clarification from us?
> > > > Looking forward to your response in the discussion period.
> > > >
> > > > Best regards,
> > > > Paper 365 Authors

---

> > > > > ### Comment · Reviewer_Gs1B · 2024-11-24
> > > > > **Response to author**
> > > > >
> > > > > >Additionally, evaluating performance would be easier with ground truth data generated from a platform like RLBench across diverse tasks.
> > > > >
> > > > > From the result provided from the rebuttal, it seems like TraceVLA and OpenVLA have very close performance, i was wondering if you guys could provide me with standard deviations and conduct statistical significance tests (e.g., p-tests) to provide more robust evidence of the results.

---

> > > > > > ### Author Response · Authors · 2024-11-25
> > > > > > **Response to Reviewer Gs1B**
> > > > > >
> > > > > > Thank you for your timely response. We would like to clarify that our original numbers were based on 50 rollouts with random initial states, with the average performance calculated across all 10 tasks per suite (4 suites in total in LIBERO). Following your suggestion, we conducted evaluations with two additional random seeds (a total of three random seeds) and included the standard deviation, which has been updated in Table 5. As outlined in our evaluation protocol, each suite involves 10 tasks with 50 rollouts per task, totaling 500 episodes per suite. With such a large number of evaluations, the standard deviation is understandably small. Nonetheless, TraceVLA consistently outperforms the baseline OpenVLA.
> > > > > >
> > > > > > While our primary focus is on learning generalist policies from real-robot data, where TraceVLA demonstrates the most significant performance gains over OpenVLA in real-world tasks, the consistent performance improvement in the LIBERO benchmark still provides strong evidence for the effectiveness of our proposed visual prompting technique.
> > > > > >
> > > > > > Thank you again for your feedback. If you have any further questions, please feel free to ask. We would be happy to address them.

---

> > > > > > > ### Comment · Reviewer_Gs1B · 2024-11-25
> > > > > > > **Thanks for the response**
> > > > > > >
> > > > > > > My concern have been addressed. I raise my rating.

---

### Official Review · Reviewer_xXSr · 2024-11-03

**Soundness:** 3
**Presentation:** 3
**Contribution:** 3
**Rating:** 8
**Confidence:** 4

**Summary:**

This paper proposed a visual trace prompting method called TraceVLA to enhance VLA models in manipulation tasks.
By incorporating Co-Tracker to visually prompt keypoint trajectories into existing VLA frameworks, TraceVLA achieves better performance than baselines in simulated and real robot experiments. It also shows better generalziation across environmental variations than vanilla VLA model and text prompted ones, further proves the advantage of the proposed visual trace method.

**Strengths:**

1. Clear description of the method. Comprehensive ablation on prompt setting and trace length, as well as analysis of memory and speed.
2. Regarding the drawback in VLA practice, this work creatively applies tracking as visual prompting for VLAs to learn the invariance across different scene settings in manipulation.

**Weaknesses:**

1. Sec 4.1 is a bit hard to read. Mixing figures, tables, and your main context together with reduced blanks is not a good idea to present your experiment results clearly.
2. Lack of details in experiments. For example, how do you define success, especially in real robot experiments? When you add noise to the environment, do you have a limit over related parameters? Is there any case study that qualitatively shows the advantage of applying visual tracking prompts? Please add them to your revision, which is also good for reproducibility.

**Questions:**

1. How does this method generalize under camera orientation changes? By nature it helps the model against illumination/background changes, but how can VLA remain effective when the camera moves, given only 2D tracking result? If the change is small, then to what degree do you vary? You mentioned in the real robot section that finetuning is needed due to domain shift, but not for the simulation. Is that also due to the domain gap being small in simulated experiments?
2. Regarding the active point selection, did you choose the $\kappa$ so that $M=5$ takes a reasonable ratio in the active points, and therefore similar traces are learned in the training set? If not, does it have the potential to generalize across visually different robots?
3. Will the color, thickness, or even transparency of the visual prompt affect the performance?
4. Is K somehow related to the size of the image patch size?
5. Will the performance drop if the action is moving into/out of the camera frame, or rotating in/out of it?

---

> ### Author Response · Authors · 2024-11-23
> **1/3 Response to Reviewer xXSr**
>
> Thank you for your insightful feedback and review! We are encouraged that you think our visual trace prompting technique is novel and our ablation study across various design choices is thorough. Below, we address your concerns and questions in detail.
>
> > Sec 4.1 is a bit hard to read. Mixing figures, tables, and your main context together with reduced blanks is not a good idea to present your experiment results clearly.
>
> Thank you for your suggestion. We have reorganized the figures, tables and texts in Section 4.1 for better readability.
>
> > Lack of details in experiments. For example, how do you define success, especially in real robot experiments? When you add noise to the environment, do you have a limit over related parameters? Is there any case study that qualitatively shows the advantage of applying visual tracking prompts? Please add them to your revision, which is also good for reproducibility.
>
> We have added comprehensive experimental details in Appendix A, including: success criteria for each task in real-robot experiments, environmental initializations and description of distracting objects, and detailed evaluation protocols for both simulation and real-world settings. Additionally, we have added a qualitative visualizations of TraceVLA against baseline OpenVLA in Appendix C.
>
> > How does this method generalize under camera orientation changes? By nature it helps the model against illumination/background changes, but how can VLA remain effective when the camera moves, given only 2D tracking result?  If the change is small, then to what degree do you vary?
>
> We would like to clarify that, in this paper, we follow the settings of existing robot generalist policies such as OpenVLA, Octo, and RT1/RT2-X. Specifically, we use a third-person camera view for observations, which is mounted and remains stationary throughout each episode.
>
> However, we would like to highlight that our visual trace prompting technique is versatile and can be extended to handle egocentric views. To address camera motion in such cases, a **homography transformation** can be applied to align historical traces with the current observed frames. This involves calculating a 3×3 homography transformation matrix between points in a historical frame (e.g., at time $t-k$) and the current frame (time $t$). Using this matrix, we can transform historical points to align with the current frame, thus enabling the generation of visual traces for moving robot arms and/or objects while compensating for camera motion.
>
> A concrete example demonstrating this approach is available [here](https://anonymous.4open.science/r/TraceVLA_ICLR2025_Rebuttal-0394/homograph_transformation.pdf), where we apply homography transformation to an egocentric video from the EpicKitchen dataset.
>
> Extending our visual trace prompting to egocentric videos, and potentially leveraging large-scale action-free human videos, is indeed an exciting direction for future work. However, this exploration falls beyond the scope of the current paper and is left as a future research direction.
>
>
> > You mentioned in the real robot section that finetuning is needed due to domain shift, but not for the simulation. Is that also due to the domain gap being small in simulated experiments?
>
> Regarding the need for finetuning in real robot experiments but not in simulation, this difference stems from broader domain gaps beyond just camera variations - real-world scenarios introduce additional challenges like lighting variations, sensor noise, and imperfect actuation that aren't present in simulation. The simulation experiments primarily test geometric viewpoint variations in a controlled environment, making the domain gap relatively smaller and manageable without additional finetuning, while real-world deployment requires bridging multiple domain differences simultaneously.

---

> > ### Author Response · Authors · 2024-11-23
> > **2/3 Response to Reviewer xXSr**
> >
> > > You mentioned in the real robot section that finetuning is needed due to domain shift, but not for the simulation. Is that also due to the domain gap being small in simulated experiments?
> >
> > Regarding the need for finetuning in real robot experiments but not in simulation, this difference stems from broader domain gaps beyond just camera variations - real-world scenarios introduce additional challenges like lighting variations, sensor noise, and imperfect actuation that aren't present in simulation. The simulation experiments primarily test geometric viewpoint variations in a controlled environment, making the domain gap relatively smaller and manageable without additional finetuning, while real-world deployment requires bridging multiple domain differences simultaneously.
> >
> >
> > > Regarding the active point selection, did you choose the \kappa so that M=5 takes a reasonable ratio in the active points, and therefore similar traces are learned in the training set? If not, does it have the potential to generalize across visually different robots?
> >
> >
> > We empirically determined the value of $\kappa$ through statistical analysis and cross-validation on the OXE training data. The objective was to filter out background points unrelated to the robot arms or manipulated objects. During model development, we observed that background points were not always entirely static due to minor camera motions or errors introduced by the CoTracker algorithm. These inaccuracies could result in misleading traces that negatively affect the model's learning process. To address this, we aimed to strike a balance between the precision and recall for active points so that they most likely imply the motions related to manipulation tasks, regardless of the trace similarities. Afterwards, we randomly select maximally M=5 active points from the candidate pool so that their traces are informative while minimizing visual interference. According to our experiments, these hyperparameters fit well across different viewpoints, emboidment and environments in the OXE datasets.

---

> > > ### Author Response · Authors · 2024-11-23
> > > **3/3 Response to Reviewer xXSr**
> > >
> > > > Will the color, thickness, or even transparency of the visual prompt affect the performance?
> > >
> > > To address your question, we conducted additional ablation studies on these visualization parameters, including fine-tuning the model on datasets with different trace variations and altering trace visualization settings. Our findings indicate that performance variations across these parameters are minimal within a reasonable range. Below is the detailed experimental results for each setting
> > >
> > > **Thickness**:
> > > | Thickness | SimplerEnv Average Success Rate|
> > > |--------|----------------|
> > > | linewidth=1 | 47.2% |
> > > | **linewidth=2** (**TraceVLA**) | 47.7% |
> > > | linewidth=3 | 47.8% |
> > >
> > > **Transparency** (Default TraceVLA uses $\alpha=1$, the lower $\alpha$ is, the more transparent of the visual trace that we add)
> > > | Transparency | SimplerEnv Average Success Rate|
> > > |--------|----------------|
> > > | $\alpha$=1 (**TraceVLA**) | 47.7% |
> > > | $\alpha$=0.8 | 47.3% |
> > >
> > > **Color**: (RYPBG stands for [Red, Yellow, Purple, Blue, Green] which is used in TraceVLA, POBGG stands for [Pink, Orange, Blue, Grey, Green])
> > > | Color | SimplerEnv Average Success Rate|
> > > |--------|----------------|
> > > | RYPBG (**TraceVLA**) | 47.7% |
> > > | POBGG | 47.3% |
> > >
> > > **Takeaways**:
> > > The choice of thickness, transparency, and color has only a negligible impact on TraceVLA's performance when chosen within reasonable ranges. These parameters do not require extensive hyperparameter tuning, simplifying the selection process. We hope this clarification addresses your concerns and highlights the robustness of TraceVLA to variations in these parameters.
> > >
> > >
> > >
> > > > Is K somehow related to the size of the image patch size?
> > >
> > > K is consistently set across all experiments and is indeed related to the image patch size. In this paper, we use images with a resolution of 256x256, following OpenVLA. Thus, no domain-specific fine-tuning of K is required. Choosing K is straightforward and intuitive: while a larger K can improve point tracking results, it also increases computational cost. To select K, we visualized visual traces from several example trajectories generated by Co-Tracker across a range of K values and chose the smallest K that reliably tracks the robot arm and active moving objects.
> > >
> > >
> > > > Will the performance drop if the action is moving into/out of the camera frame, or rotating in/out of it?
> > >
> > >
> > > As long as the robot arm or object remains partially within the camera frame, CoTracker can reliably track visible moving parts and generate visual traces, enabling the model to leverage historical motion information. Even under partial occlusions, CoTracker, as a robust point-tracking algorithm, can often maintain tracking, ensuring meaningful visual traces even when parts of the robot arm temporarily move out of view.
> > > Here, we provide three qualitative examples from our training dataset where, despite the end-effector moving in and out of the scene, CoTracker annotations still yield reasonable visual trace prompting: [example 1](https://anonymous.4open.science/r/TraceVLA_ICLR2025_Rebuttal-0394/trace_0.png), [example 2](https://anonymous.4open.science/r/TraceVLA_ICLR2025_Rebuttal-0394/trace_1.png), [example 3](https://anonymous.4open.science/r/TraceVLA_ICLR2025_Rebuttal-0394/trace_2.png).
> > > However, if the robot arm consistently moves completely out of the camera frame for extended periods, additional strategies may be required. These could include conditioning the model on historical observation data or optimizing camera placement to ensure better workspace coverage.
> > >
> > >
> > >
> > >
> > > Thank you again for your time and effort in reviewing our paper! We sincerely hope our responses have adequately addressed your concerns and look forward to any further discussion. Please let us know if you have any further questions. We are more than happy to address them.

---

> > > > ### Author Response · Authors · 2024-11-26
> > > > **Does our response address your concerns?**
> > > >
> > > > Dear Reviewer xXSr,
> > > >
> > > > Thank you so much for your time and effort in reviewing our work and providing insightful feedback. As the discussion phase is drawing to a close, we kindly request that you review our responses and confirm whether they adequately address your concerns.
> > > >
> > > > We welcome any further discussions and value the opportunity for continued improvement of our work.
> > > >
> > > > Best regards,
> > > >
> > > > Paper 365 Authors

---

> > > > > ### Comment · Reviewer_xXSr · 2024-11-27
> > > > >
> > > > > My concerns are addressed. I'll raise my rating.

---

### Author Response · Authors · 2024-11-23
**General Response**

## Review Highlights

We sincerely thank all reviewers for their thorough feedback and constructive suggestions. We particularly appreciate the acknowledgment of:

* Novelty of our proposed visual trace prompting method (**Reviewer Gs1B, 4XPZ**)
* Clear presentation and methodology (**Reviewers xXSr, cSN9, vJK6**)
* Technical contributions including efficient methodology and our proposed dataset (**Reviewers xXSr, Gs1B, 4XPZ, qxt2**)
* Comprehensive empirical validation demonstrating strong generalization (**Reviewers Gs1B, 4XPZ, qxt2**)

## Main Concerns Addressed

* Generalization Capacity (Reviewers xXSr, Gs1B, 4XPZ):
We extended evaluation on LIBERO benchmark and additional validation on out-of-domain real-robot tasks with novel objects, goals, language instructions, and motion scenarios, further highlighting the substantial generalization improvement from visual trace prompting across diverse simulation and real-world environments
* Trace Quality and Tracking Robustness (Reviewers xXSr, 4XPZ):
We conducted additional ablation studies on visualization parameters show TraceVLA maintains consistent performance under various trace conditions and illustrated its robustness in imperfect tracking scenarios.

These responses address key concerns while providing new experimental results that strengthen our paper's contributions. We incorporate our clarifications and additional analyses in our revised manuscript.

## Summary of paper updates
Main Text:
* Reorganized **Section 4.1** for better readability
* Enhanced related work discussion with newly suggested references (**Section 6**)

Appendix:
* Detailed real-robot experimental setup and protocols (**Appendix A**)
* Out-of-domain real-robot experiments to further demonstrate generalization (**Appendix B**)
* Qualitative analysis of real-robot rollouts (**Appendix C**)
* Ablation studies on trace visualization parameters (**Appendix D**)
* Comparative study with OpenVLA using varying historical observations (**Appendix E**)

These updates are highlighted in red for easy reference. We believe our work makes significant contributions to the ICLR robotics community, further strengthened by the reviewers' constructive feedback.

We sincerely thank the reviewers and AC for their thoughtful comments during the discussion period. We hope our responses have adequately addressed all concerns and welcome any further discussion.

---

### Meta-Review · Area_Chair_2Epo · 2024-12-24

**Metareview:**

The paper introduces TraceVLA, a vision-language-action model enhanced with visual trace prompting to improve spatial-temporal awareness in robotic manipulation, achieving state-of-the-art performance and robust generalization across simulated and real-world tasks with improved inference efficiency.

All reviewers acknowledge the contributions of this work, highlighting its (1) novelty, (2) technical advancements, including the efficient methodology and proposed dataset, (3) clear presentation, and (4) comprehensive empirical evaluation.

The authors have also effectively addressed the reviewers' concerns during the discussion period. All reviewers are in unanimous agreement to accept this paper.

**Additional Comments On Reviewer Discussion:**

Since the reviewers were in unanimous agreement to accept this paper, no significant discussion took place during the Reviewer Discussion phase.

---

### Decision · Program_Chairs · 2025-01-22

Accept (Poster)